# A natural WNT signaling variant potently synergizes with *Cdkn2ab* loss in skin carcinogenesis

Paul Krimpenfort[1,2], Margriet Snoek[2], Jan-Paul Lambooij[1,2], Ji-Ying Song[3], Robin van der Weide[1,4], Rajith Bhaskaran[1,2], Hans Teunissen[1,4], David J. Adams[5], Elzo de Wit [1,4] & Anton Berns[1,2]

*Cdkn2ab* knockout mice, generated from 129P2 ES cells develop skin carcinomas. Here we show that the incidence of these carcinomas drops gradually in the course of backcrossing to the FVB/N background. Microsatellite analyses indicate that this cancer phenotype is linked to a 20 Mb region of 129P2 chromosome 15 harboring the *Wnt7b* gene, which is preferentially expressed from the 129P2 allele in skin carcinomas and derived cell lines. ChIPseq analysis shows enrichment of H3K27-Ac, a mark for active enhancers, in the 5′ region of the *Wnt7b* 129P2 gene. The *Wnt7b* 129P2 allele appears sufficient to cause in vitro transformation of *Cdkn2ab*-deficient cell lines primarily through CDK6 activation. These results point to a critical role of the *Cdkn2ab* locus in keeping the oncogenic potential of physiological levels of WNT signaling in check and illustrate that GWAS-based searches for cancer predisposing allelic variants can be enhanced by including defined somatically acquired lesions as an additional input.

[1] Oncode Institute, The Netherlands Cancer Institute, Plesmanlaan 121, 1066 CX Amsterdam, The Netherlands. [2] Division of Molecular Genetics, The Netherlands Cancer Institute, Plesmanlaan 121, 1066 CX Amsterdam, The Netherlands. [3] Department of Experimental Animal Pathology, The Netherlands Cancer Institute, Plesmanlaan 121, 1066 CX Amsterdam, The Netherlands. [4] Division of Gene Regulation, The Netherlands Cancer Institute, Plesmanlaan 121, 1066 CX Amsterdam, The Netherlands. [5] Wellcome Trust Sanger Institute, Wellcome Genome Campus, Cambridge CB10 1SA, UK. Correspondence and requests for materials should be addressed to A.B. (email: a.berns@nki.nl)

The *CDKN2ab* locus on chromosome 9p21 in human (chromosome 4 in mouse) harbors the genes for three tumour suppressor proteins: p16[INK4a] and p14[ARF] (p19[Arf] in mice) encoded by *CDKN2a* and p15[INK4b] by *CDKN2b* (ref. [1]). Both p15[INK4b] and p16[INK4a] are able to induce cell cycle arrest in G1 by inhibiting cyclin-dependent kinases CDK4 and CDK6 thereby relieving the cell cycle suppression through the retinoblastoma (RB1) family of tumour suppressor proteins. The unrelated p14[ARF] protein acts primarily through MDM2 to activate the key checkpoint protein TRP53, thereby inducing either cell cycle arrest (both in G1 and in G2) or apoptosis. The entire locus is frequently lost in a wide range of primary tumours.

Previously, we have described the generation *Cdkn2ab* knockout mice derived from 129P2 ES cell clones. The first generations of *Cdkn2ab* knockout mice on a mixed 129P2;FVB/N background frequently developed skin tumours and various soft tissue sarcomas, tumour types not normally seen in *Cdkn2a*[−/−] mice[2]. However, the skin tumour phenotype was gradually lost upon backcrossing the knockout *Cdkn2ab* allele onto the FVB/N background suggesting the involvement of one or more 129P2 alleles in addition to *Cdkn2ab* loss in skin tumour development.

In this study to gain insight into this phenomenon we repeated the procedure to generate *Cdkn2ab* knockout mice, but for the backcrossing to FVB/N we now selected skin tumour bearing mice to enable the identification of potentially relevant 129P2 loci involved by analysing the 129P2 contribution in the progeny with and without the skin tumour phenotype. Microsatellite analyses indicate that the skin cancer phenotype is linked to a 20 Mb region of 129P2 chromosome 15 harbouring the *Wnt7b* gene. The

results show that the *Cdkn2ab* locus fulfils a critical role in keeping the oncogenic potential of physiological levels of WNT signalling in check. In addition, our study illustrates that the identification of cancer predisposing alleles in genome-wide association studies might be facilitated by focussing on tumours carrying distinct somatically-acquired oncogenic lesions[3].

## Results

**Cdkn2ab knockout mice develop tumours.** The search for the relevant 129P2 allele(s) involved in the skin cancer phenotype of *Cdkn2ab* KO mice was started by generating chimeric mice using the 129P2 ES cell clone carrying knockout alleles for p15[Ink4b] and p16[Ink4a] and conditionally knockout for p19[Arf] (ref. [2]). This compound mutant allele, which we called *Ink4ab*[-], enables Cre-mediated inactivation of p19[Arf] thereby generating a mutant *Cdkn2ab* locus for all three tumour suppressor proteins (Supplementary Figure 1). Chimeric mice generated from *Ink4ab*[-] ES cells were crossed to FVB/N mice harbouring a germline-expressing Cre transgene (*Act*-Cre) to obtain F1 129P2;FVB/N mice carrying either an *Ink4ab*[-] allele or a *Cdkn2ab*[-] allele. Subsequently, these mice were intercrossed to obtain a cohort of homozygous *Cdkn2ab* knockout mice unable to express any of these three tumour suppressor proteins, on a mixed 129P2;FVB/N background.

Within 4 months, many *Cdkn2ab*[−/−] animals develop multiple skin tumours in addition to other tumour types. Per animal up to 15 skin lesions develop synchronously all over the trunk (Fig. 1a) and are first detectable by palpation at the age of 8 weeks. The

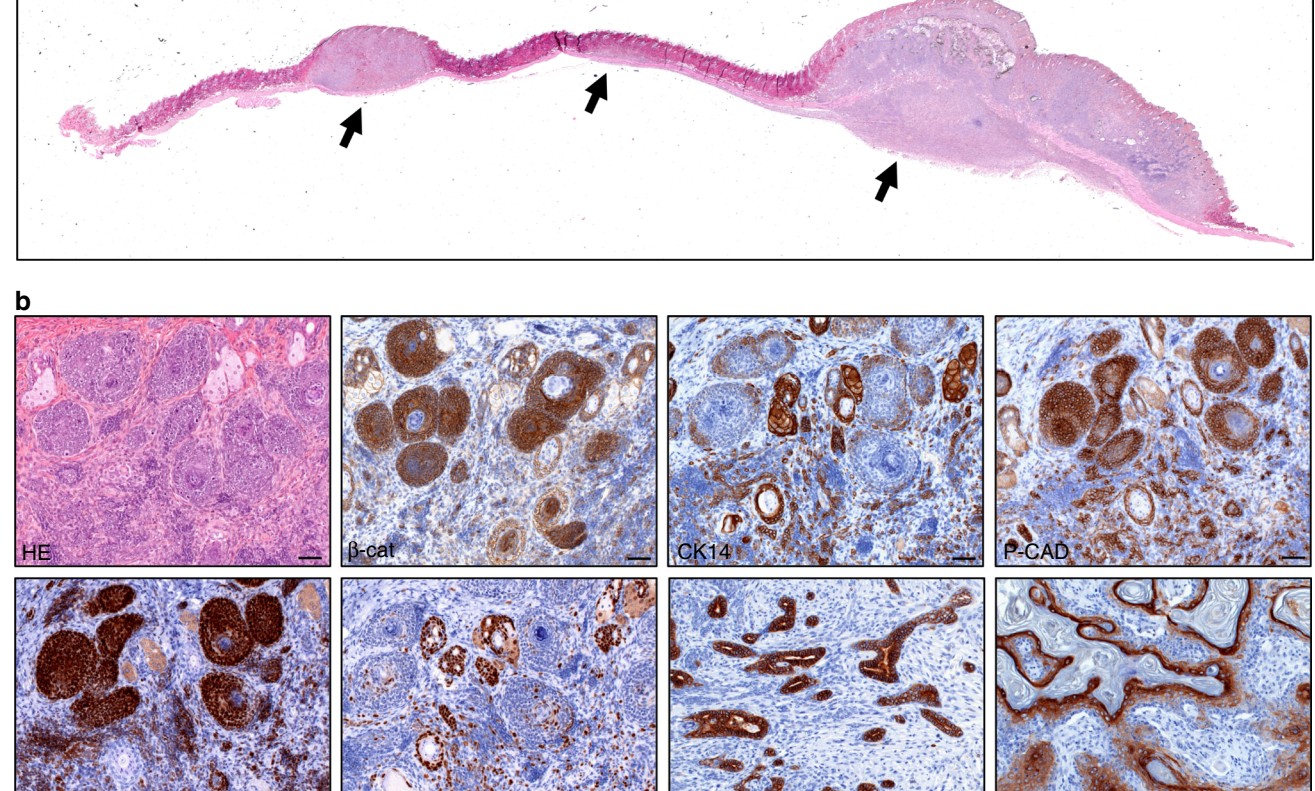

**Fig. 1** *Cdkn2ab*[−/−] mice show multiple skin trichoblastic carcinomas. **a** Total scan image of the skin region representing multiple skin tumours (arrows, scale bar 100 mm). **b** H&E and IHC characterization of skin tumours consisting largely of hair germ (LEF1+, P-CAD+, SOX9+), basal/squamous (CK14+/CK10+) cells and dermal papillary cells (LEF1+), also showing strong regional β-Catenin positivity (scale bars: 20 μm)

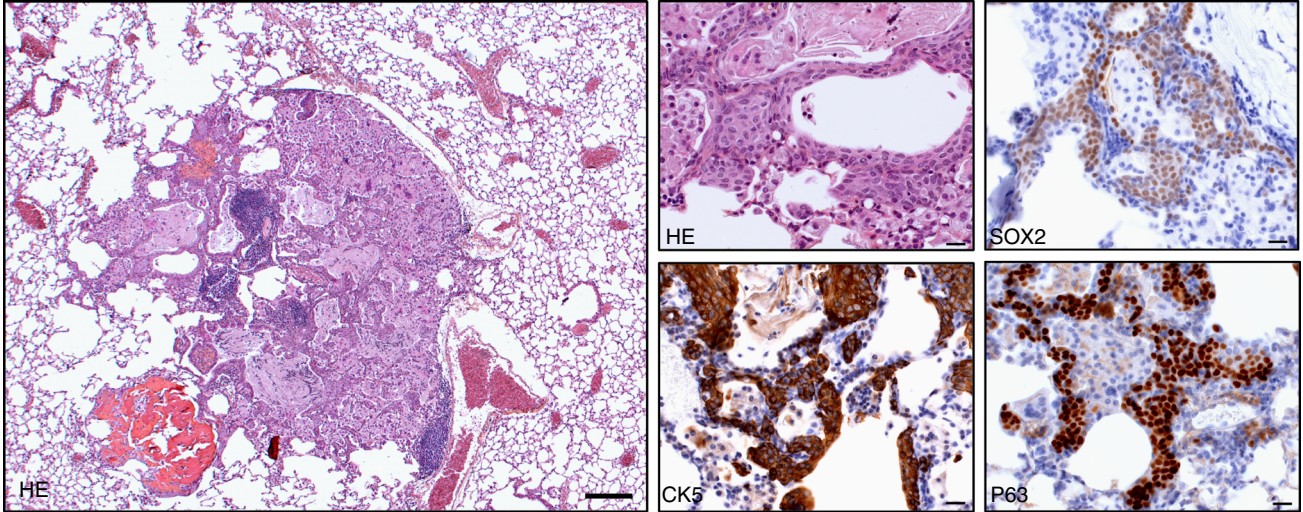

**Fig. 2** *Cdkn2ab*$^{-/-}$ mice show squamous lung metaplasia. H&E (scale bars 200 μm and 20 μm) and CK5, SOX2 and P63 IHC (scale bar: 20 μm)

tumours frequently show biphasic morphology and are often composed of different cell types, likely reflecting the oncogenic transformation of an early progenitor. Conventional H&E staining and immunohistochemistry (IHC) show that the tumours consisted largely of hair germ, basal and squamous cells and dermal papillary cells and that they form hair bulb-like structures with densely packed strands and small nodules (Fig. 1b). Based on these characteristics the tumours are diagnosed as trichoblastic carcinomas (TBCs) that originate from early hair follicle progenitor cells[4–6]. Strong regional LEF1 expression and occasional nuclear β-catenin positivity point to canonical WNT cascade activation (Fig. 1b). Interestingly, in 20% of the skin tumour bearing *Cdkn2ab*$^{-/-}$ mice local squamous metaplasia (Fig. 2) in the bronchiolar as well as alveolar compartments of the lung (19 out of 91) is observed and in only 1% *Cdkn2ab*$^{-/-}$ mice without skin tumours (1 out of 95, $p < 0.005$),

The stochastic skin tumour development suggests that additional somatic (epi-)genetic events, or specific local environmental conditions, are required for tumour induction. The latter is supported by the frequent occurrence of local hypertrophic hair follicles in *Cdkn2ab*$^{-/-}$ mice in the telogen phase skin before the onset of skin tumours (Supplementary Figure 2). Exome sequencing of 23 independent skin tumours using tail DNA as matched controls does not reveal any recurrent mutations in coding regions. However, copy number analysis reveals frequent chromosome losses (e.g., chr.14) as well as the presence of small amplicons or modest gain of chromosomes, e.g., chr. 8, 11, and 15 (Supplementary Figure 3). This might explain the stochastic nature of skin tumour onset when (some of the) chromosomal gains can enhance the expression of oncogenic drivers such as *c-Myc* as has been shown for the gain of chromosome 15 in a number of mouse models[7].

**The Chr 15: 85 MB region synergises with Cdkn2ab loss**. To identify the relevant 129P2 allele(s) in affected mice we performed a genome-wide scan using 134 microsatellite markers selected to be polymorphic between 129P2 and FVB/N with an average coverage of ~20 Mb (~10 cM) (for information on microsatellites see Supplementary Table 1). Of a group of 38 *Cdkn2ab*$^{-/-}$ mice of the first generations, 13 mice developed skin tumours with strong genetic linkage to a marker on Chromosome

15 (D15Mit107 at 85 Mb). Further backcrossing of skin tumour bearing *Act-Cre;Cdkn2ab*$^{-/-}$ mice to *Ink4ab*$^{-/-}$ mice (FVB/N background) supported the linkage to this chromosomal segment and subsequent fine mapping identifies an interval of 20 Mb on Chromosome 15 (Chr15: 67–87 Mb) (Fig. 3a) with genetic linkage to the skin carcinoma phenotype. No *Cdkn2ab*$^{-/-}$ mice homozygous for the FVB allele at Chr15: 85 Mb region developed skin carcinoma's ($n = 95$), whereas the penetrance of the skin carcinoma phenotype in mice heterozygous for 129P2 allele at Chr15: 85 Mb is over 75% (70 out of 91) (Fig. 3b, c). Likewise, these data also support the contribution of *Cdkn2ab* loss and the 129P2 Chr15: 85 Mb region to aberrant squamous differentiation in the lung (Fig. 3d) since these lesions are observed almost exclusively in skin tumour bearing mice. In contrast, the other tumour types are observed in *Cdkn2ab*$^{-/-}$ mice with or without the 129P2 Chr 15: 85MB allele (Fig. 3c).

Mouse strains carrying *Cdkn2a* (ref. [8–10]) or *Cdkn2b* (ref. [11]) mutant alleles published previously do not develop skin tumours. However, the cohorts analysed in these studies did not carry the 129P2 Chr15: 85 Mb region. To determine whether the loss of all 3 tumour suppressors encoded by the *Cdkn2ab* locus is required for skin carcinogenesis we generated 3 additional cohorts of mice carrying the relevant 129P2 region but each heterozygous knockout for either one of the three *Cdkn2ab* encoded proteins while being full knockout for the other 2 (see Supplementary Figure 1 for schematics of all *Cdkn2ab* mutant alleles used in this study). Skin tumours develop in a small fraction (10%) of *Cdkn2a* knockout mice hemizygous for *Cdkn2b*, however, with a much longer latency than full *Cdkn2ab* knockout mice (Fig. 3e). No skin tumours are seen in mice heterozygous for modified alleles still encoding either p16$^{Ink4a}$ or p19$^{Arf}$ only. These data indicate that each of the three *Cdkn2ab* encoded proteins can effectively suppress skin carcinogenesis in mice carrying the 129P2 Chr15: 85 Mb region.

The Chr15: 67–87 Mb region harbours, among many other genes (see Supplementary Table 2), the genes encoding *Wnt7b* and *Pdgfß*. Both genes are functional in putative oncogenic signalling pathways and fulfil a critical role in the development of hair follicles[12,13] and lung[14,15]. In addition, p15$^{Ink4b}$ is highly expressed[16,17] during the development of these tissues. Therefore, we decided to further explore the relevance of these genes in combination with the complete *Cdkn2ab* loss in skin carcinomas and lung lesions. mRNA

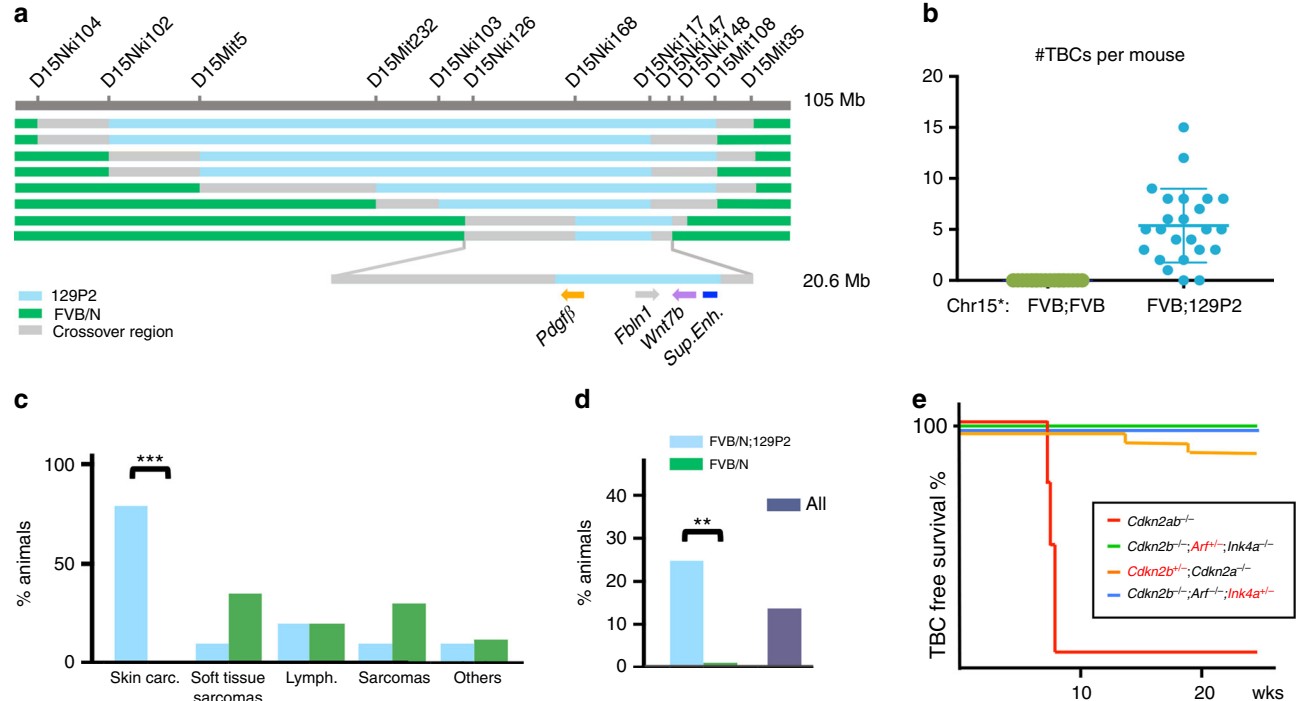

**Fig. 3** Identification of the Chr15 79–87 Mb region having genetic linkage to the skin tumour phenotype. **a** Schematic of the recombined Chr15 with varying 129P2 and FVB/N background in skin tumour bearing mice (blue: 129P2, green: FVB/N background and grey: region of crossover); relevant microsatellite markers used are indicated on top; below a blowup of the relevant region). **b** Number of skin tumours on individual *Cdkn2ab⁻/⁻* mice homozygous FVB/N (green) Chr15 85 Mb or FVB/N;129P2 (blue) for region Chr 15 85 Mb (Chr15*). **c** Tumour spectrum of *Cdkn2ab⁻/⁻* mice as mentioned in **b**. **d** Frequency of squamous metaplasia in the lungs of *Cdkn2ab⁻/⁻* mice as mentioned in **b**. **e** Kaplan–Meier curve showing skin tumour-free survival of *Cdkn2ab mutant* mice FVB/N;129P2 for the Chr 15; 85 Mb region; red line: *Cdkn2ab⁻/⁻*: deficient for p15$^{Ink4b}$, p16$^{Ink4a}$ and p19$^{Arf}$; green: *Cdkn2b⁻/⁻;Arf⁺/⁻;Ink4a⁻/⁻*: deficient for p15$^{Ink4b}$ and p16$^{Ink4a}$, proficient for p19$^{Arf}$; blue: *Cdkn2b⁻/⁻;Arf⁻/⁻;Ink4a⁺/⁻*: deficient for p15$^{Ink4b}$ and p19$^{Arf}$, proficient for p16$^{Ink4a}$; orange: *Cdkn2b⁺/⁻; Cdkn2a⁻/⁻*: deficient for p16$^{Ink4a}$ and p19$^{Arf}$, proficient for p15$^{Ink4b}$. Statistical analysis (Unpaired *t*-test): ns: **$p < 0.01$, ***$p < 0.001$

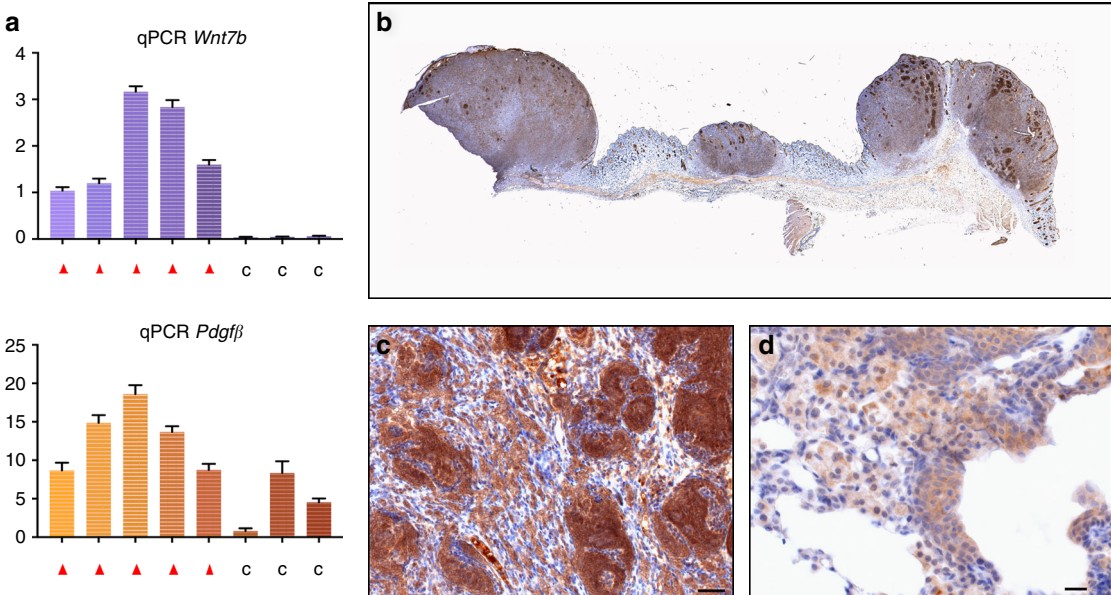

**Fig. 4** WNT7b and PDGFβ expression in skin tumours and lung metaplasias. **a** mRNA (qPCR) analysis of *Wnt7b* and *Pdgfβ* (relative to control) of 5 independent skin tumours (red triangle) and three soft tissue sarcomas (c). **b** Total scan image (scale bar: 100 mm), and **c** high power image of WNT7b IHC in primary skin tumours (scale bar: 20 µm). **d** High power image of WNT7b IHC in lung metaplasia (scale bar: 20 µm). Statistical analysis (Unpaired *t*-test): ns: not significant, *$p < 0.05$, **$p < 0.01$, ***$p < 0.001$. Error bars represent standard deviation of the mean (3 replicates)

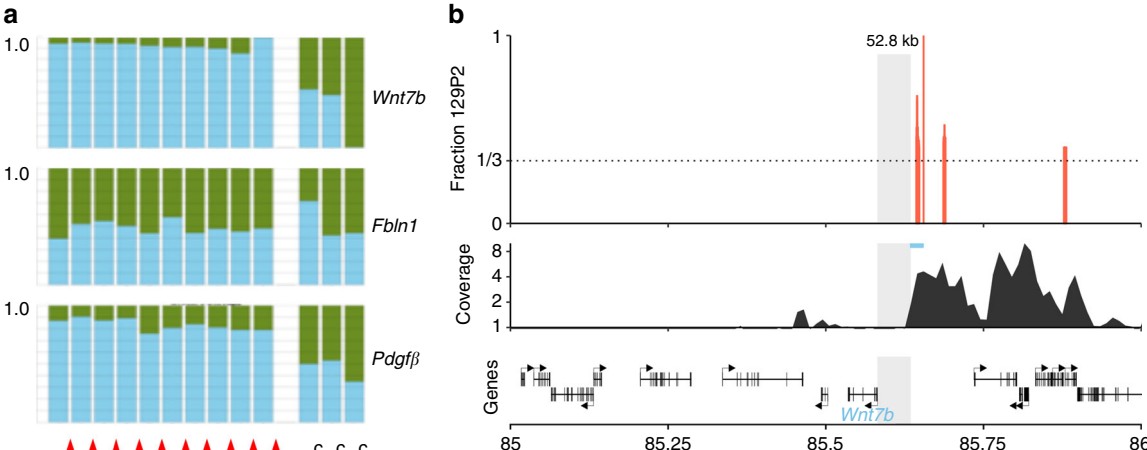

**Fig. 5** The implication of 129P2 Wnt7b allele in the development of skin tumours. **a** Fraction (*Y*-axis) of FVB/N allele derived transcripts (green) and 129P2 allele derived transcripts (blue) of *Wnt7b*, *Fbln1* and *Pdgfβ* in 10 independent skin tumours and 3 control (soft tissue sarcomas) tumours (*X*-axis). **b** Window of region of interest (chr15:85644260-85645382) with significant enrichment of the 129P2 allele reads in the H3K27-Ac ChIP-seq data region of interest. Non-informative (no SNP differences between FVB/N and 129P2) reads and reads without significant difference between FVB/N and 129P2 are not indicated. The super-enhancer region is 52.8 kb upstream of the transcription start site of *Wnt7b* (blue)

analysis and immunohistochemistry show that *Wnt7b* is expressed in the skin carcinomas of *Cdkn2ab*$^{-/-}$ mice (Fig. 4a–c) but not in soft tissue sarcomas. WNT7b expression is also prominent in the squamous lesions in the lungs of these animals (Fig. 4d). *Pdgfβ* is expressed in the skin tumours as well, but also in the soft tissue sarcomas (Fig. 4a). Allele-specific mRNA analyses based on 3' UTR polymorphisms (see Methods section) shows that in the skin carcinomas both genes are transcribed primarily from the 129P2 alleles, explaining the dependence of skin carcinoma development on the presence of this allele (Fig. 5a). Notably, the *Fibulin1* gene located in between the *Pdgfβ* and *Wnt7b* genes does not show allele-specific expression. Since the amino acid sequences of both PDGFβ and WNT7b encoded by FVB/N and 129P2 genes are identical this suggests that variation in their regulatory sequences or post-transcriptional mRNA stability underlies the enhanced expression of the 129P2 alleles in the skin tumours. To assess whether the 129P2 region has an altered chromatin landscape, we derived tumour cell clones able to grow in soft agar from three independent skin tumours. In these clones *Wnt7b* expression is reduced and *Pdgfβ* expression is hardly detectable (Supplementary Figure 4a). However, as observed in the primary tumours, in these cell clones the *Wnt7b* gene is transcribed with a similar allele specificity although to a lesser extent (Supplementary Figure 4b). We performed H3K27-Ac ChIP-seq analysis on three clones derived from independent tumours to identify active enhancers. Of note, the cell lines analysed shows a gain of FVB/N chromosome 15, leading to an average allelic balance of 33 percent 129P2 instead of 50 percent. The allele-specific analysis identifies four regions with a significant increase (i.e. higher than the expected allelic balance) of H3K27-Ac on the 129P2 allele (Supplementary Figure 5). One of these regions, chr15: 85643259–85648162, is juxtaposed upstream of a known super-enhancer mentioned in the dbSUPER database (https://www.ncbi.nlm.nih.gov/pubmed/26438538) (Fig. 5b). This region is located 52 kb upstream of the *Wnt7b* promoter and has a 2-fold higher fraction of the 129P2 allele (FDR < 1%). The relative increase of H3K27-Ac in this super-enhancer region on the 129P2 allele is consistent with the allele preference of *Wnt7b* transcriptional activity. There are multiple polymorphisms in this super-enhancer region also affecting several transcription factor binding

motifs. Further investigation is needed to identify the relevant motifs responsible for the here observed differential transcriptional control of *Wnt7b*.

**Transformation by Wnt signalling is mediated by Cdk6.** To gain insight into the underlying mechanism of the cooperation between enhanced *Wnt7b* and *Pdgfβ* expression and loss of the *Cdkn2ab* locus we turned to defined cell culture systems: MEFs and tumour-derived cell lines. We used MEF's from *Cdkn2a*$^{-/-}$ and *Cdkn2ab*$^{-/-}$ mice not carrying the 129P2 Chr15: 67–87 Mb region. Since WNT7b acts in the canonical WNT signalling pathway, we used the well-defined WNT3a conditioned medium as a source of WNT protein to assay their responsiveness to WNT signalling. We transfected a WNT-responsive Luciferase reporter construct in *Cdkn2a*$^{-/-}$ and *Cdkn2ab*$^{-/-}$ MEF's and show that β-catenin dependent transcription is strongly induced by WNT3a supplementation in *Cdkn2a*$^{-/-}$ and *Cdkn2ab*$^{-/-}$ MEF's to comparable levels (Supplementary Figure 6a). A control Luciferase reporter construct not containing *Wnt* responsive elements is hardly expressed in *Cdkn2a*$^{-/-}$ and *Cdkn2ab*$^{-/-}$ MEF's with or without WNT3a supplementation. However, *Cdkn2ab*$^{-/-}$ MEF's shows a much stronger proliferative response to WNT stimulation than *Cdkn2a*$^{-/-}$ MEF's in adherent cultures (Supplementary Figure 6b). In contrast, PDGFβ does not affect proliferation of *Cdkn2ab*$^{-/-}$ MEF's with or without WNT3a supplementation (Supplementary Figure 7). Using rich 10% serum HITES medium (see Methods) WNT supplementation also strongly promotes colony formation and growth in soft agar of *Cdkn2ab*$^{-/-}$ but not of *Cdkn2a*$^{-/-}$ MEF's (Fig. 6a). These results indicate that, at least in MEFs, p15$^{Ink4b}$ restricts the effects of canonical WNT signalling.

Next, we turned to the cell clones derived from the *Cdkn2ab*$^{-/-}$ skin tumours and focused on unique features associated with transformation i.e., anchorage-independent growth and migratory capacity. Under regular adherent culture conditions supplying additional PDGFβ does not change the proliferation of these tumour cell clones nor does it affect their anchorage-independent growth (Supplementary figure 8). WNT3a conditioned medium has no substantial effect on the growth of tumour cell clones under adherent conditions but

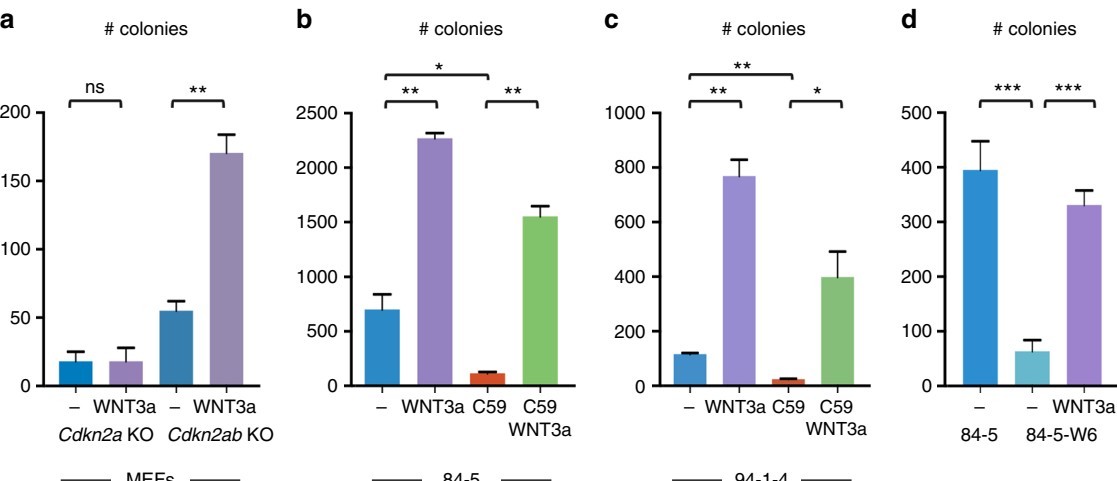

**Fig. 6** Validation of functional involvement of WNT7b in the transformed nature of MEFs and skin tumour cell lines. **a** Anchorage-independent colony outgrowth of *Cdkn2a* KO and *Cdkn2ab* KO MEFs with and without WNT3a supplementation. **b** Anchorage-independent colony formation of tumour cell lines 84-5 under regular (−) conditions, in the presence of WNT3a supplementation, in the presence of the porcupine inhibitor C59, in the presence of C59 and WNT3a supplementation. **c** Anchorage-independent colony formation of tumour cell line 94-1-4 under regular (−) conditions, in the presence of WNT3a supplementation, in the presence of the porcupine inhibitor C59, in the presence of C59 and WNT3a supplementation. **d** Anchorage-independent colony formation of tumour cell lines 84-5 and 84-5-W6 knockout for *Wnt7b* with and without Wnt3a supplementation. Statistical analysis (Unpaired *t*-test): ns: not significant, *$p < 0.05$, **$p < 0.01$, ***$p < 0.001$. Error bars represent standard error of the mean (3 biological replicates)

significantly stimulates the proliferation in soft agar conditions although this response does vary between tumour cell clones (Supplementary figure 9 and Fig. 6b, c). These data indicate that WNT signalling is critical for the anchorage-independent colony formation of tumour cells and this raises the question to what extent the WNT ligands produced by the tumour cells themselves are determining their growth behaviour. To test this, we used the C59 Porcupine inhibitor to block WNT palmitoylation and thereby its function[18]. Addition of C59 prevents their anchorage-independent outgrowth, which can be rescued by WNT3a conditioned medium (Fig. 6b, c). However, since C59 affects the function of all WNTs this experiment did not show that the elevated expression of WNT7b is required for soft agar growth. Therefore, we disrupted the *Wnt7b* gene in skin tumour cells using CRISPR/Cas9 technology. Inactivation of the *Wnt7b* gene blocks the soft agar outgrowth of tumour cells to the same extent as addition of C59 and again this can be largely rescued by WNT3a conditioned medium (Fig. 6d and Supplementary figure 10). Evidence for CRISPR/Cas *Wnt7b* gene inactivation is shown by the significantly reduced transcription of the canonical WNT signalling target *Axin2* in edited tumour cells which can be rescued by WNT3a supplementation (Supplementary figure 11). In line with these observations, shRNA knockdown of *Wnt7b* leads to a reduction of soft agar colony outgrowth and the impairment correlates with the extent of knockdown (Supplementary figure 12). A similar dependence on *Wnt7b* is found for the migratory activity of the skin tumour cell clones as *Wnt7b* inactivation impairs the migration of tumour cells in a conventional scratch assay (Supplementary figure 13). All together, these results indicate that in the chr15: 67–87 Mb region *Wnt7b* is a critical gene for the transformation of *Cdkn2ab*−/− skin tumour cells at least in vitro. We note, however, that enhanced *Pdgfβ* expression might contribute to tumour development in vivo as growth factors and/or nutrients are likely less abundantly available in vivo.

Next, we asked how WNT signalling might contribute to the transformation of *Cdkn2ab*−/− cells and focused on the role of *Cdk6/CyclinD* for the following reasons: first, CDK6/CyclinD is inhibited by the *Cdkn2ab*-encoded Ink4 proteins; second,

both WNT and CDK6/CyclinD are involved in transformation-related features such as anchorage-independent growth and migration[19–24]; third, WNT signalling enhances both *CyclinD* and *Cdk6* transcription through upregulation of *Myc* (ref. [25]). Indeed, qPCR assays show that WNT signalling enhanced *CyclinD* transcription in both *Cdkn2a* KO and *Cdkn2ab* KO MEFs (Supplementary figure 14) and transcription of both *Cdk6* and *CyclinD* in the skin carcinoma cell lines (Fig. 7a and Supplementary figure 14). In addition, a reduction of *Cdk6* transcripts is observed in *Wnt7b* knockout tumour cells, which can be restored by WNT3a supplementation (Fig. 7a). ShRNA knockdown of *Cdk6* expression does not affect proliferation of *Cdkn2ab*-deficient MEFs and skin tumour cell lines under adherent conditions (data not shown) but does reduce the anchorage-independent colony outgrowth, which correlates with the extent of knockdown (Supplementary figure 15, 16). To independently assess the role of *Cdk6* we generated *Cdk6* knockout tumour cell clones using CRISPR/Cas9 gene editing (Supplementary figure 17). These clones show very limited anchorage-independent colony outgrowth capacity, which can be only marginally rescued by WNT3a supplementation (Fig. 7b). Loss of *Cdk6* also affects the migratory capacity of the tumour cell clones (Supplementary figure 13). Overexpression of CDK6 by retroviral transduction (Supplementary figure 17) results in very efficient soft agar colony outgrowth both of skin tumour cell lines, with or without inactivation of the endogenous *Wnt7b* allele, and *Cdkn2ab* KO MEFs (Fig. 7c, d). To test whether phosphorylation of pRB pocket proteins per se is critical for anchorage-independent growth we also transduced constitutive active CDK4-R24C into *Cdkn2ab* KO MEFs but its overexpression does not lead to increased soft agar growth capacity (Fig. 7d) pointing to a specific role for CDK6 in cellular transformation.

## Discussion
Our data indicate that the contribution of WNT signalling to cellular transformation is mediated to a significant extent by stimulating CDK6 activity. Our observations fit a scenario in which p15[Ink4b] plays a critical role in the hair follicle growth cycle

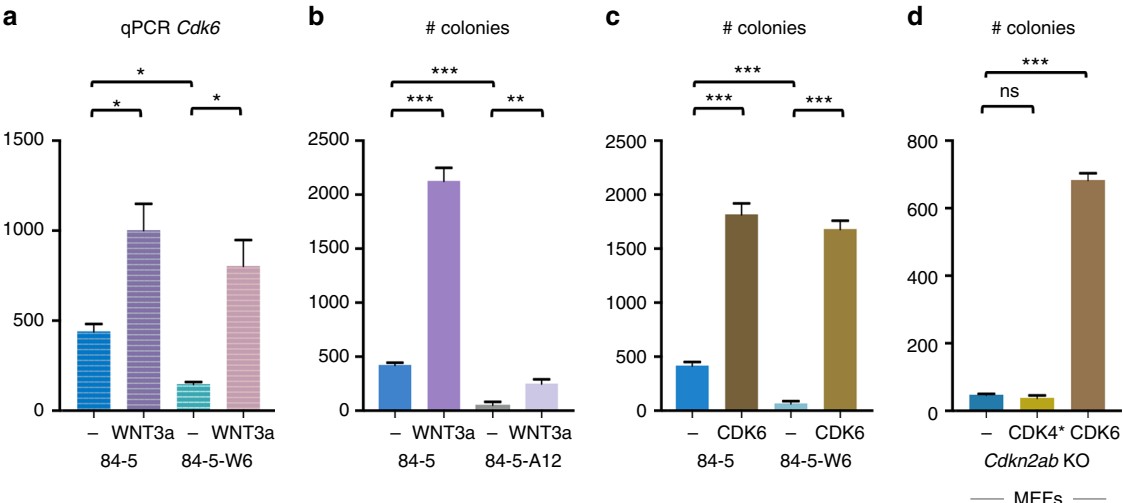

**Fig. 7** Validation of functional involvement of CDK6 in WNT induced transformation of MEFs and skin tumour cell lines. **a** mRNA analyses (qPCR) of *Cdk6* relative to *Gapdh* of 84-5 tumour cells and of 84-5-W6 tumour cells knockout for *Wnt7b* with and without WNT3a supplementation. **b** Anchorage-independent colony formation of tumour cell lines 84-5 and 84-5-A12 (*Cdk6* knockout derivative) with and without WNT3a supplementation. **c** Anchorage-independent colony formation of tumour cell lines 84-5 and 84-5-W6 (*Wnt7b* knockout derivative) with and without overexpression of retrovirally transduced *Cdk6*. **d** Anchorage-independent colony formation of *Cdkn2ab*$^{-/-}$ MEFs with and without overexpression of retrovirally transduced *Cdk6* and *CDK4-R24C* (CDK4*). Statistical analysis (Unpaired *t* test): ns: not significant, *$p < 0.05$, **$p < 0.01$, ***$p < 0.001$. Error bars represent Standard Deviation of the Mean (3 replicates) (**a**) or Standard Error of the mean (3 replicates) (**b–d**). *Cdkn2ab* knockout mice develop skin tumours but this phenotype is lost on backcrossing of the mice. Here, the authors describe a genetic variant encompassing *Wnt7b* that synergises with *Cdkn2ab* loss and is required for tumour formation in these mice

by keeping CDK6 activation through WNT signalling in check. In the absence of p16$^{Inka}$ and p19$^{Arf}$ this then creates a favourable setting for tumourigenesis.

Unfortunately, graft experiments of the early passage tumour-derived cell lines in immunodeficient NSG mice did not result in tumour outgrowth as is often observed with primary tumours making it impossible to confirm the effects of WNT7b in this complementary in vivo setting. However, our data are fully in line with a number of published observations. Upregulation of WNT7b has been reported in hfSCs during physiological and precocious anagen after BMP inhibition[12]. Furthermore, *Wnt7b* conditional gene targeting during HF morphogenesis disrupts HF cycling causing a shorter[12] anagen, premature catagen onset with overall shorter hair production. These 'in vivo' observations strongly support the stimulating effect of natural WNT7b overexpression on hfSCs and thereby its oncogenic role in the development TBCs. In addition, clinical data also implicate a contribution of subtle modulation of WNT7b expression to the development of oral and oesophageal squamous carcinoma[26,27].

It is important to emphasize that p15$^{Ink4b}$ is constitutively expressed in the hair follicle, kidney and lung, this in contrast to p16$^{Ink4a}$. Especially in conditions of high levels of WNT ligands such as is the case for WNT7b in the hair follicle, CDK6 activation in the absence of p15$^{Ink4b}$ could lead to increased anchorage-independence, migration and proliferation of progenitor cells. The absence of a backup of p16$^{Ink4a}$ or p19$^{Arf}$ to stop cells that respond to these oncogenic signals would increase the risk of accumulating genomic aberrations leading to tumour development.

Loss of *Cdkn2b* could pose a similar risk in other physiological conditions with allelic variation in WNT signalling and/or CDK6 activation itself. This might apply to kidney development which involves several WNT family members[28] with concurrent high expression of p15$^{Ink4b}$ (ref. [17]). Of note, renal cell carcinomas frequently show inactivating mutations in *CDKN2b* (ref. [29]) concomitant with WNT pathway activation[30]. Intriguingly, allelic variants of DKK3, which is implicated in WNT signalling control, have been associated with predisposition to renal cell carcinoma[31]

and squamous lung carcinoma[32] as well. Furthermore, both p15$^{Ink4b}$ and CDK6 are critically implicated in the regulation of hematopoietic stem cell fate in mouse and man[33–35]. Interestingly, loss of *CDKN2b* expression[36] and enhanced transcription of natural *CDK6* alleles are very frequently observed in several leukaemia's such as AML and ALL (ref. [34]). Therefore, *CDKN2b* deficiency and variation in CDK6 activation could put progenitor cells at risk to become pre-malignant.

Our studies also point to the significance of the deletion of the complete *CDKN2ab* locus in specific tumour subsets, such as mesothelioma, squamous cell carcinoma, and glioblastoma making it worthwhile to further explore whether WNT and/or CDK6 plays a critical role in these tumours even in the absence of any mutations in these genes. This could motivate a much broader therapeutic application of specific CDK6 and WNT pathway inhibitors.

In conclusion, we here have described a cancer predisposing QTL that poses a very high risk when co-occurring with the frequently observed somatic deletion of the *Cdkn2ab* tumour suppressor locus. Put in a broader perspective, this might well be a recurrent theme in tumour development with a critical role for WNT signalling to maintain stem cells involved in tissue renewal and regeneration. Their proliferation has been recently coined as a risk factor for cancer[37]. If the level of WNT signalling is indeed such a critical component of tumour predisposition, it might be easily missed in GWA studies since next to the large number of *WNT* genes there are many components in the WNT signalling pathway that can influence this very signalling cascade and consequently many alleles might confer such risk. It would require quantitative downstream readouts to assess whether the risks conveyed by variant alleles acting in this pathway might be held responsible for some of the unresolved cancer predispositions. Focussing GWA studies on cancer patient subgroups with somatic disruptions of the *CDKN2AB* locus could serve as a starting point for identifying such alleles. It might be also worth exploring if inhibiting WNT and/or CDK6 signalling is of benefit to these patients.

## Methods

**Animal experiments.** All animal experiments comply with international regulations and ethical guidelines and have been authorized by the experimental animal committee at The Netherlands Cancer Institute.

**Cdkn2ab mutant alleles.** The construction of the *Ink4ab−* allele (knockout for p15[Ink4b] and p16[Ink4a]) enabing the Cre mediated derivation of the *Cdkn2ab−* allele (knockout for p15[Ink4b], p19[Arf] and p16[Ink4a]) (ref. [2]) and the conditional *Cdkn2a* allele enabling the Cre mediated derivation of the *Cdkn2a−* (knockout for p16[Ink4a] and p19[Arf]) were described previously[2,10]. The *Cdkn2b−;Arf[F]* mutant allele (knockout for p15[Ink4b] and conditional knockout for p19[Arf]) enabling the Cre mediated derivation of the *Cdkn2b−;Arf−* allele (knockout for p15[Ink4b] and knockout for p19[Arf]) was generated by replacing the *Cdkn2b* exon 1 and 2 by a hygromycin expression cassette carrying a 3' LoxP site followed by the insertion of a LoxP site 3' of exon 1β of *Cdkn2a*. These modifications were performed in 129P2 ES cells using conventional targeting procedures. Mice carrying knockout alleles for *Cdkn2ab*, *Cdkn2a* and *Cdkn2b;Arf* were obtained by crossing *Act-cre* mice expressing Cre in the germline (ref. [2]) to *Ink4ab−*, *Cdkn2a[F]* and *Cdkn2b−;Arf[F]*, respectively. Additional information on the alleles can be found on the MGI website, accession numbers: allele *Ink4ab−*: 2384165, *Cdkn2ab−*: 3767459, *Cdkn2a[F]*: 2384163, *Cdkn2b;Arf[F]*: 5560507.

**Histological analysis.** Tissues were fixed in either 4% formalin or acidified formaldehyde (ethanol/acetic acid/formaldehyde/saline at 40:5:10:45 v/v). Sections were prepared at 2 µm thickness from the paraffin blocks and stained with haematoxylin and eosin (HE) according to standard procedures. For immunohistochemistry (IHC), 4 um-thick sections were made. Antibodies used in this study are listed in Supplementary Table 4.

Antibody staining was revealed using either diaminobenzidine or 3-amino-9-ethylcarbazole chromogen. Images were captured using a Zeiss Axioskop2 Plus microscope (Carl Zeiss Microscopy) equipped with a Zeiss AxioCam HRc digital camera, and processed using AxioVision 4 software (Carl Zeiss Vision).

**DNA profiling.** DNA profiling of the back crossed mice was performed by a genome-wide scan (autosomes) using a panel of 134 polymorphic microsatellite markers distinguishing allele length between 129P2 and FVB genotypes at a mean density of ~20 Mb (~10 cM). For fine mapping on chromosome 15 an additional set of 8 polymorphic markers was used. Primers around the microsatellites were used to PCR the short DNA intervals, followed by agarose gel electrophoresis (4% gels), length polymorphism was visualized by ethidium bromide staining using a UV transilluminator. List of satellite markers and related primers see Supplementary Table 1.

**Exome sequencing.** Exonic DNA was captured using SureSelect Mouse All Exon capture baits. Captured material was indexed and sequenced on the Illumina Hiseq platform at the Wellcome Trust Sanger Institute. Raw 75 bp pair-end sequencing reads were aligned with BWA to the mouse reference genome (mm10) prior to somatic variant calling.

**Copy number variation analysis.** DNA copy number profiles were generated from whole-exome sequencing data using CopywriteR which uses off-target reads from the exome capture[38].

**Chipseq.** Chromatin immune precipitations were performed as described[39] with slight modifications. Cell lines were grown until 70% confluency and For one experiment 10 million cells were crosslinked for 10 min in 1% formaldehyde after it is stopped by adding Glycin (0.1 M end concentration). After washing the cells for 2× with PBS by centrifugation for 5 min at 1350 G, lyses is started in 10 ml lyses buffer 1 (50 mM HEPES pH 7.5, 140 mM NaCl, 1 mM EDTA, 10% Glycerol, 0.5% NP40, 0.25% Triton X-100) for 30 min at 4 °C. Cells were centrifuged for 5 min at 1350G to remove lyses buffer 1. The cell pellet was lysed for 10 min in lyses buffer 2 (200 mM NaCl, 1 mM EDTA, 0.5 mM EGTA, 10 mM Tris-HCl). After removing lyses buffer 2 by centrifugation the pellet was incubated for 30 min in lyses buffer 3 (100 mM NaCl, 1 mM EDTA, 0.5 mM EGTA, 10 mM Tris-HCl ph 8, 0.1% DOC, 0.5% N-Lauroyl sarcosine). The DNA of 3 million cells was sheared on a covaris ultrasonicator in a covaris microtube AFA (ref 520045) under the following conditions: peak power 75 W, duty cycle 10%, cycle/burst 200 for 7 min. For every sample the shearing was performed three times. 50 µl Dynabeads protG magnetic beads (Invitrogen 10003D) were blocked in PBS with 0.5 BSA for every chip. 5 µg antibody was coupled to the beads for 3 h. Following a washing step with lysis buffer 3 the beads were resuspended in 500 µl lyses buffer 3 containing 1% Triton X-100. The cell extract was added to the beads and incubated for 12 h. 4 Washing steps were performed with RIPA buffer (50 mM HEPES pH 7.5, 1 mM EDTA, 0.7% DOC, 1% NP-40, 0.5 M LiCl) and one with cold TE, 50 mM NaCl. Elution was done over night with 150 µl elution buffer (50 mM Tris-HCl ph 8, 10 mM EDTA, 1% SDS) at 65 °C. For the DNA recovery the Qiaquick pcr purification kit (Qiagen 28106) was used. Libraries were prepared with the KAPA HTP library preparation kit (KK8234 Illumina).

Products were sequenced using an Illumina HiSeq 2500 (single end 65 bp). Reads were mapped to GRCm38 using bowtie2 with default settings. The main

analysis was performed by counting the number of ChIP-seq reads supporting either the 129P2 or FVB/N allele, obtained from the Mouse Genomes Project[40]. Using a sliding window of variable size (250-1000 bp), we summed the counts for each allele and tested for a deviance[41] from the expected ratio (set as the median allelic ratio per chromosome) with an exact Poisson test. The Fisher's combined probability test was used to combine the P-values of the three clones, followed by the Benjamini–Hochberg procedure. All windows with a FDR < 0.05 were selected.

**Cell culture.** Tumour cell lines were derived from primary cultures of chopped tumour pieces; after 1 week of primary culture in 10%, HITES medium, suspensions of $25 \times 10^3$ single cells were cultured in soft agar to select cell clones able to proliferate anchorage-independently. Clones were picked and maintained under adherent conditions for further experiments. Cells were grown at 37 °C, 5% $CO_2$, either in DMEM (GIBCO) supplemented with 10% FCS, 100 units per ml penicillin, 100 mg ml⁻¹ streptomycin or in DMEM/F12 (GIBCO) supplemented with 10% FCS, 100 units per ml penicillin, 100 mg ml⁻¹ streptomycin, and HITES (insulin, transferrin, selenium, hydrocortison, EGF). Porcupine inhibitor C59 was obtained from Cellagen Technology (C7641-10) and used at a final concentration of 10 nM. PDGFβ was obtained from: WNT3a conditioned medium was supplemented to regular medium in a 1:3 ratio. MEFs from E13.5 embryos were isolated and cultivated according to standard protocols. For soft agar growth analysis, cells (24,000 cells per well in a 6-well plate) were resuspended in 0.4% soft agar and plated on a bottom layer of 1% agar. The number of colonies growing in soft agar after two weeks of culture was quantified using ImagePro Plus 5.1.2 software (MediaCybernetics)

**β-Catenin dependent transcription assay.** Activation of the WNT/β-catenin pathway in *Cdkn2a⁻/⁻* and *Cdkn2ab⁻/⁻* MEFs was measured using a TOP-FOP assay. One day prior to transient transfection, *Cdkn2a⁻/⁻* and *Cdkn2ab⁻/⁻* MEFs were seeded in a density of 50.000 cell per 6-well. The TOP-WRE (Wnt-responsive element) reporter plasmid or FOP- MRE (Mutant WNT-responsive element) reporter plasmid were transfected transiently into the MEFs using Lipofectamine (Lipofectamine LTX; Invitrogen art#15338-100). After 24 h the medium was replaced by L1 + Wnt3a conditioned medium and for a negative control L1 medium without Wnt3a. 24 h after adding Wnt3a conditioned medium, luciferase activity was measured using the Dual Luciferase assay from Promega (cat# E4530) and normalized with the co-transfected renilla reporter

**Protein expression analysis.** Cell pellets were lysed in RIPA buffer and the lysates were cleared by centrifugation at 20,800×g for 10 min. Proteins were separated on 4–12% NuPAGE Novex Bis-Tris gels (Invitrogen), transferred to nitrocellulose (Schleicher and Schuel) or Immobilon (Millipore) membrane and probed with antibodies. Immuno-complexes were visualized by enhanced chemo-illuminescence (Pierce 34080) using goat anti-mouse (AMI3404, Biosource) or goat anti-rabbit (ALI0404, Biosource) antibodies coupled to horseradish peroxidase. The primary antibodies used were: WNT7b: Abcam (ab94915); CDK6: Thermo (AHZ0232).

**RNA expression analysis.** RNA was isolated from frozen tumour pieces and cell pellets using RNEasy columns (Qiagen). cDNA was prepared using the Tetro cDNA synthese kit from Bioline (bio-65043). qPCR was performed using Sensifast Sybr (Hi-rox) kit purchased from Bioline (BIO92020) with the following primer sets: Wnt7b: exon 3 frw: TCGACTTTTCTCGTCGCTTT, exon 4 rev: AGG CTTCTGGTAGCTGCGTA; Pdgfβ: exon 5 frw: TGGGACATCCAGGGAG CA, exon 6 rev: CCTTGTCATGGGTGTGCTTA; Fbln1: exon 7 frw: AGCTGTG GGACTGGCTATG, exon 9 rev: TGCTCTTGGGATGGCTG; Axin2 exon 5 frw: CAAGACCAAGGAGGAGATCG, exon 6 rev: TTTTGGCAAGGTACCACCTC; Cdk6: exon 7 frw: CTCAACCCATCGAGAAGTTTG, exon 8 rev: GTTGGAT GGCAGGTGAGAGT; CyclinD1: exon 3 frw: TGGTGAACAAGCTCAAGTGG, exon4 rev: GCAGGAGAGGAAGTTGTTGG; Gapdh: exon 4 frw: GCCGGCTCA TCACACAGT, exon 4 rev: TCGGAGTCCTCAGTCTCACTC; Actin: exon 4 frw: CACAACGTGCCCATCTATGA, exon 5 rev: GGCCATCTCATTCTCGAAGT. Samples were analysed at least in triplicate. Reactions were performed using the ABI Prism 7000 sequence detection system.

**Analysis of allele-specific expression.** RNA isolation and cDNA preparation was performed as described above. Wnt7b mRNAs from 129P2 and FVB/N differ in the 3' UTR at position 505 in exon 4 (G in 129P2 and C in FVB/N, creating a Nar1 restriction site in FVB/N). Part of exon 4 was PCR amplified from cDNA using primers around nucleotide position 505 of exon 4 (frw: AACACGCACCAGTAC ACCAA, rev: GCTCCAGAAGCAAAGAAGGA). PCR products were sequenced and scored for G or C at position 505 of exon 4. Pdgfβ mRNAs from 129P2 and FVB/N differ in the 3' UTR at position 176 of exon 7 (G in 129P2 and C in FVB/N, creating a Nsp1 restriction site in FVB/N). Exon 5–7 was PCR amplified from cDNA using primers located in exon 5 (frw: TGGGACATCCAGGGAGCA) and downstream of position 176 of exon 7 (rev: TCGGGCTAAGTGCCAGGCT). PCR products were sequenced and scored for G or C at position 176 of exon 7. Fbln1 mRNAs from 129P2 and FVB/N differ in the 3' UTR at position 172 of exon 15 (C in 129P2 and A in FVB/N, creating a Nco1 restriction site in FVB/N). The relevant region of exon 15 was PCR amplified from cDNA using primers located in exon 15 (frw: AGGAGGGCTTTTTCACCACT) and downstream of nucleotide position

172 of exon 15 (rev: TGCTCTTGGGATGGCTG). PCR products were sequenced and scored for C or A at position 172 of exon 15.

**CRISPR/Cas Gene editing**. Tumour cell lines were modified according to standard procedures involving transfections of pX330 vectors expressing Cas9 and specific guideRNAs (gRNA) (ref. [39]). Transfections were done using Lipofectamine LTX from Invitrogen (15338–100). gRNA designs were obtained from the Crispr Design website from MIT (http://crispr.mit.edu/). After transient selection for puromycin resistance clones were picked, expanded and molecularly characterized. The *Wnt7b* gene was inactivated by co-transfection of two pX330 vectors expressing gRNAs enabling the excision of exon 3. Sequence gRNA1: GGTGCA ATCCACTTGCTGTA, sequence gRNA2: CGGGCAGAAAGGTAGCCGTT. Editing was confimed by PCR analysis using primers around the deleted exon 3. The *Pdgfβ* gene was inactivated by co-transfection of two pX330 expressing gRNAs enabling the excision exon 2. Editing was confimed by PCR analysis using primers around the deleted exon 2. gRNA1: TCCTGGTGTCTGACCGACAG, sequence gRNA2: AGGAGGGTCCGATTTACCTA. The *Cdk6* gene was inactivated by co-transfection of two pX330 expressing gRNA enabling the excision of part of exon 3. Sequence gRNA1: GCCCGCGACCTGAAGAACGG, sequence gRNA2: GACCT TCGAGCACCCCAACG. Editing was confirmed by PCR analysis using primers around the deleted part of exon 3.

**ShRNA knockdown**. Lentiviral constructs (pLKO.1 vector) expressing shRNAs against Wnt7b and Cdk6 were obtained from The RNAi Consortium (Broad Institute/Harvard). Plasmid preparation, virus production and virus infection were performed according published protocols. Sequence of shRNA: Wnt7b-4: GCT ACCTAAGTTCCGCGAGGT; sequence of shRNA Wnt7b-7: CGGAGCATTGTC ATCCGTGGT; sequence of shRNA Cdk6–1: CCGGTTGCATCTTTGCAGAAA; sequence of shRNA Cdk6-2: ATCTTCTAGAGATAACTACTT. Supplementary Table 3 provides the sequences of primers used in the experiments described in the Methods section above.

## Data availability

The exome sequencing data of skin tumours and its matched controls have been deposited in the NCBI Sequence Read Archive with accession code PRJEB8102. Chipseq data are available from the Gene Expression Omnibus under accession code GSE125885. The authors declare that the data supporting the findings of this study are available within the paper and its Supplementary information files or available from the authors upon request.

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

## Acknowledgements

We thank Rahmen Bin Ali for technical assistance in generating the mice. We thank Colin Pritchard and other members of the Mouse Clinic for Cancer and Aging (MCCA)

Transgenic Core Facility, Oscar Krijgsman and Koen Schipper, The Netherlands Cancer Institute, for advice. We thank Jan van der Vliet, Miranda Cozijnsen and members of the animal facility of the Netherlands Cancer Institute for maintaining the mice. We thank members of the Department of Experimental Animal Pathology and members of the Sequencing Facility of the Netherlands Cancer Institute. This work was supported the Queen Wilhelmina Prize from the Dutch Cancer Society to A.B, by the European Research Council under the European Union's Seventh Framework Programme (FP7/2007-2013)/ERC synergy grant agreement n° 319661 COMBATCANCER of which A.B. is one of the principal investigators, and a National Roadmap grant for Large-Scale Research Facilities of the Netherlands Organization for Scientific Research.

## Author contributions

M.S. performed and analysed microsatellite mapping; J-P.L. was involved in the tissue culture and biochemical experiments; J-Y.S. was responsible for the histopathological analysis; R.B. and D.J.A. were involved in the exome sequencing and CNV analysis; R.v. W., H.T. and E.d.W. performed and analysed the Chipseq experiments; P.K. was responsible for the design and execution of the experiments; P.K. and A.B. wrote the paper.

## Additional information

**Competing interests:** The authors declare no competing interests.

