## [Peer Review File · Nature Communications]

Reviewers' comments:

Reviewer #1 (Remarks to the Author):

In this manuscript Krimpenfort et al investigate their observation that skin cancer incidence in *Cdkn2ab*^{-/-} mice is different on the 129 versus FVB/N strain backgrounds. This is an overall interesting observation. The effect of strain on cancer model phenotypes is rarely investigated and yet has the potential to uncover meaning molecular drivers of carcinogenesis. They show that the skin cancer promoting region maps to a certain interval on Ch 15 and provide evidence the *Wnt7b* is at least one of the drivers of this phenotype. The reason why *Wnt7b* is more highly expressed in one strain remains unclear. The authors propose a mechanism relating *Wnt7b* to *Cdk6* levels and provide additional function evidence in cell lines that change in *Cdk6* elicit similar phenotypes.

Major points:

1. While the text is quite clear, many figures are more confusing than they need to be. Some figures lack labeling on the axes. What is the point of doing such nice work then not putting any effort into the representation of the data in the Figures?
2. There is some over interpretation of the data. Ex. Figure 3a, b, e, f do not show that *Wnt7b* is "highly" expressed, only that it is expressed. The allele specific PCR is very nice and does show that the 129 allele is more highly expressed than the FVB/N allele.
3. Data confirm KO of *Wnt7b* and *Cdk6* in the CRISPR lines is missing. Over expression of *Cdk6* should be confirmed by western. Many of these types of experiments where the authors confirm their reagents are missing. I won't list them all but the author need to maintain a certain level of rigor.

Minor points:

1. Somewhere in the figures that authors should show all the genes that are within their candidate region on Ch15, rather than just states that there are lots of genes including *Wnt7b* and *Pdgbf*.
2. All IHC and H&Es need a scale bars. Fig 1f also needs a scale.
3. The use of *Wnt3a* conditioned media instead of *Wnt7b* should be better justified. Given the other data on cells with *Wnt7b* deletion (+/- *Wnt3* rescue) as well as the data on *Cdk6* this is not a question of whether they have sufficient support for more model, but rather that is just seems weird and should be justified better. Even better would be to have at least a little data with *Wnt7b*.
4. They are allowed more than 4 figures and their points could benefit from having more main text figures that show more of the gain and loss of function data on *Wnt7b* and *Cdk6*.

Reviewer #2 (Remarks to the Author):

This manuscript by Krimpenfort et al characterizes an effect of varying genetic background on mouse tumor phenotype. Such effects have been seen frequently in mouse genetic studies, but rarely characterized in any detail. The authors noted that mice carrying a segment of chr.15 derived from the 129 parental strain had a strong effect on skin tumor development in *Cdkn2ab* knockout mice. Further breeding narrowed the region to an interval of around 20Mb, and the investigation was focused on *Wnt7b*, which lies within this interval between 85-86Mb on chr.15. The authors go on to investigate the functional links between *Wnt7b* and *Cdkn2ab* knockout, and these data demonstrate convincingly that *Wnt* signaling affects the growth of cell lines derived from the skin tumors. They also nicely demonstrate that this effect may operate through *Cdk6*, although the finer details of this mechanism are not clear.

A caveat in these studies (which also applies to many other studies of mouse tumor QTLs) is that the specific polymorphism, and indeed the specific causative gene, is not definitively identified. This would require making a knock-in mouse strain in which the 129 allele is transferred to the FVB background or vice versa, and while this is possible, it has to be acknowledged that such studies are painstaking and very time-consuming. A point that could be discussed in more detail is why the authors specifically picked Wnt7b. As they acknowledge, there are many other genes in the region and some have documented effects in skin keratinocytes (eg Let7b is very close to Wnt7b and can inhibit skin wound healing Wu Y, et al. *Exp Dermatol*, 2017 Feb. PMID 27513293). Other neighboring genes may also have important functions in skin.

The authors make the point that the expression of the 129 allele is higher in tumors or tumor cell lines from heterozygous mice, and that this is possibly linked to a polymorphic super-enhancer upstream from Wnt7b. Is this also the case for normal tissue from these same animals? In Fig 3, analysis of expression is focused on tumor RNA/protein. If Wnt7b is the major germline susceptibility gene, it might be expected to have an effect on allele-specific expression that is seen before tumors develop, and that increases the probability of tumor initiation. Otherwise it is possible that the allelic differences seen only occur after tumor initiation (possibly in the context of copy number gains on chr 15) , and that the polymorphisms in the critical gene(s) are important at late rather than early stages of tumor development. The authors could discuss the possibility that observed allelic differences may only be seen after tumors arise and have begun to progress.

Minor points:

1. The resolution of the images in Fig 1 (also Fig 4 and Supp Figure 2) is not very clear and it is not obvious that the staining is really nuclear for B-cat. The scale bar should also be more prominent. It should be possible to provide high resolution images that show the histology in greater detail.

2. In fig 1F the chromosomes labelled 20 and 21 are presumably X and Y
Were any gender differences observed in tumor incidence?

3. Fig. 3 suggests that there is preferential allele-specific expression of Wnt7b and Pdgfb in the tumors, but in the cell lines (Supp Fig 3) for Pdgfb the directionality is absent or even reversed. What might be the explanation of this result, given that the cell lines have an apparent gain of the FVB alleles on chr.15 which would be expected to lead to increased expression of the FVB Pdgfb allele??

4. Fig legend 3 is very sparse and needs more detail

5. Fig 4f has no error bars

6. More information could be provided on the basis for the increased binding at the Wnt7b FVB promoter region. Are there polymorphisms that may affect binding of transcription factors in this region? Is there any evidence in favor of the conclusion that increased binding in this region is causally linked to increased expression Wnt7b?

7. Supp Fig 4. Which genes are in the regions neighboring the peaks in blue? The authors focus on the specific peak upstream from Wnt7b, but these other peaks also show allele-specific binding in favor of the FVB allele and could contribute to the phenotypes?

Reviewer #3 (Remarks to the Author):

This study deals with the involvement of Wnt signaling in skin cancer development induced by the loss of Cdkn2ab tumor suppressor locus in mice, depending on the genetic background. The authors showed that Cdkn2ab^{-/-} mice, generated from 129P2 ES cells, develop TBC-like skin carcinomas but not in the FVB/N background. They found that a 20 Mb region of 129P2 chromosome 15 harboring the Wnt7b and Pdgfb genes is linked to the skin carcinoma phenotype.

Wnt7b was shown to be overexpressed in skin carcinoma induced by the loss of Cdkn2ab and involved in oncogenic potentials of Cdkn2ab deficient cell lines in vitro. Furthermore, the authors identified Cdk6 as a downstream gene of Wnt7b signaling and Cdk6 caused in vitro transformation of skin tumor cell lines.

These studies are important for the understanding of tumorigenesis involving the somatic deletion of the Cdkn2ab tumor suppressor locus. However, phenotypes and findings of this study are observed in mice with specific genetic background and it is unclear whether author's model is consistent with tumorigenesis in human.

Major comments:

1. Importance of Wnt7b gene in skin carcinogenesis in Cdkn2ab knockout (KO) mice is not confirmed in vivo. Knockin analysis of Wnt7b in FVB/N mice would support their conclusions. Analysis of the effects of Wnt7b KO on skin carcinogenesis in 129P2 background is also useful. At least subcutaneous and orthotopic tumor models using Cdkn2ab deficient carcinoma cell lines with or without Wnt7b could show the importance of Wnt7b in vivo.

2. It is not clear that single Cdkn2a KO cells do not respond to Wnt3a in the cell proliferation and colony formation assays (Fig.4b and c). The authors state that upregulation of Cdk6 activity rather than its amount induced by the complete loss of Cdkn2ab locus is important for Wnt-induced cellular transformation. To support their conclusion, the expression levels of Cdk6 in Cdkn2a KO cells (MEFs) treated with or without Wnt3a should be shown compared with those of Cdkn2ab KO cells. In addition, the phosphorylation level of the substrate for Cdk6, such as Rb, needs to be shown to assess the Cdk6 activity in the same condition.

3. In Fig. 4f, the amount of β -catenin or the expression of Axin2 or SP5, Wnt target genes, should be shown to assess the change of endogenous Wnt/ β -catenin signaling activity.

4. Statistical analyses are lacking in this study. Statistical significance should be shown in the individual data.

Minor comments:

1. Label and explain the axes in Fig.1e, 1f, and 3a-3c.

2. Fig.3a does not have a control group.

3. In page 6, line 12, "WNT antagonist DKK3" is an inappropriate description. It is reported that DKK3 does not have Wnt antagonist activity.

We thank the reviewers for their critical and constructive remarks. We have addressed the issues raised adapted the text and figures accordingly. We improved the organization and quality of the figures in the following aspects: 1) the resolution of the histological pictures; 2) we present now more main figures each of them focusing on a more defined item; 3) statistical significance of the data is given in each panel when appropriate; 4) to improve the clarity of the story line some of the data in the main figures are now presented as suppl. figures and the other way around. Control tumours in figures 4 and 5 represent soft tissue sarcoma's in *Cdkn2ab* deficient mice.

Reviewers' comments:

Reviewer #1 (Remarks to the Author):

In this manuscript Krimpenfort et al investigate their observation that skin cancer incidence in *Cdkn2ab*^{-/-} mice is different on the 129 versus FVB/N strain backgrounds. This is an overall interesting observation. The effect of strain on cancer model phenotypes is rarely investigated and yet has the potential to uncover meaning molecular drivers of carcinogenesis. They show that the skin cancer promoting region maps to a certain interval on Ch 15 and provide evidence the *Wnt7b* is at least one of the drivers of this phenotype. The reason why *Wnt7b* is more highly expressed in one strain remains unclear. The authors propose a mechanism relating *Wnt7b* to *Cdk6* levels and provide additional function evidence in cell lines that change in *Cdk6* elicit similar phenotypes.

Major points:

1. While the text is quite clear, many figures are more confusing than they need to be. Some figures lack labeling on the axes. What is the point of doing such nice work then not putting any effort into the representation of the data in the Figures?

See general remarks above

2. There is some over interpretation of the data. Ex. Figure 3a, b, e, f do not show that *Wnt7b* is “highly” expressed, only that it is expressed. The allele specific PCR is very nice and does show that the 129 allele is more highly expressed than the FVB/N allele.

Indeed, the data in figure 3a (now figure 4a) only show that these genes are expressed; we now include control tumours and left out the western blot. Control tumours in figures 4 and 5 represent soft tissue sarcoma's in *Cdkn2ab* deficient mice.

3. Data confirm KO of *Wnt7b* and *Cdk6* in the CRISPR lines is missing. Over expression of *Cdk6* should be confirmed by western. Many of these types of experiments where the authors confirm their reagents are missing. I won't list them all but the author need to maintain a certain level of rigor.

Validation of *CDK6* inactivation and overexpression has been done by Western blots and presented in suppl. figure 17. Confirmation of inactivation of *Wnt7b* in cell lines by western blot is complicated due to secretion and instability of the protein. Inactivation of *Wnt7b* was based on Crispr/Cas mediated the excision of exon 3, which leads to an out of frame fusion between exon 2 and 4. Homozygous excision of exon 3 was confirmed by PCR using primers surrounding exon 3. The loss expression of the *Wnt* signaling target *Axin2*, which can be rescued by *WNT3a* supplementation, serve in our view as convincing evidence for *Wnt7b* gene inactivation.

Minor points:

1. Somewhere in the figures that authors should show all the genes that are within their candidate region on Ch15, rather than just states that there are lots of genes including *Wnt7b* and *Pdgfb*.

We now present in suppl. table 2 all the genes in the relevant region including the position of the micro satellite markers; in addition we indicate whether genes are possibly implicated in carcinogenesis one way or the other; this is based on literature data.

2. All IHC and H&Es need a scale bars. Fig 1f also needs a scale.

All histology figures now include scale bars.

3. The use of *Wnt3a* conditioned media instead of *Wnt7b* should be better justified. Given the other data on cells with *Wnt7b* deletion (+/- *Wnt3* rescue) as well as the data on *Cdk6* this is not a question of

whether they have sufficient support for more model, but rather that it just seems weird and should be justified better. Even better would be to have at least a little data with Wnt7b.

We have used WNT3a conditioned medium for WNT supplementation of regular media. This has been and still is a widely used standard procedure. The commercially available WNT7b preparations have not been validated.

4. They are allowed more than 4 figures and their points could benefit from having more main text figures that show more of the gain and loss of function data on Wnt7b and Cdk6.

We present now more main figures each of them focusing on a distinct aspect which we hope improves the story line of the manuscript.

Reviewer #2 (Remarks to the Author):

This manuscript by Krimpenfort et al characterizes an effect of varying genetic background on mouse tumor phenotype. Such effects have been seen frequently in mouse genetic studies, but rarely characterized in any detail. The authors noted that mice carrying a segment of chr.15 derived from the 129 parental strain had a strong effect on skin tumor development in Cdkn2ab knockout mice. Further breeding narrowed the region to an interval of around 20Mb, and the investigation was focused on Wnt7b, which lies within this interval between 85-86Mb on chr.15. The authors go on to investigate the functional links between Wnt7b and Cdkn2ab knockout, and these data demonstrate convincingly that Wnt signaling affects the growth of cell lines derived from the skin tumors. They also nicely demonstrate that this effect may operate through Cdk6, although the finer details of this mechanism are not clear.

A caveat in these studies (which also applies to many other studies of mouse tumor QTLs) is that the specific polymorphism, and indeed the specific causative gene, is not definitively identified. This would require making a knock-in mouse strain in which the 129 allele is transferred to the FVB background or vice versa, and while this is possible, it has to be acknowledged that such studies are painstaking and very time-consuming. A point that could be discussed in more detail is why the authors specifically picked Wnt7b. As they acknowledge, there are many other genes in the region and some have documented effects in skin keratinocytes (eg *Let7b* is very close to *Wnt7b* and can inhibit skin wound healing Wu Y, et al. *Exp Dermatol*, 2017 Feb. PMID 27513293). Other neighboring genes may also have important functions in skin.

Indeed, many other genes in the identified regions could contribute to the skin tumorigenesis. We concentrated on *Wnt7b* and *Pdgfb* because they are implicated in skin and lung development and might function in oncogenic signaling. This holds true also for the *Let7* family members *Let7-b* and *Let7-c* (both are located in close vicinity on the other side of the relevant super enhancer region. However, there are no sequence differences in *Let7-b* and *Let7-c* between FVB/N and 129P2, which makes it impossible to analyze allele preferences for these genes.

The authors make the point that the expression of the 129 allele is higher in tumors or tumor cell lines from heterozygous mice, and that this is possibly linked to a polymorphic super-enhancer upstream from *Wnt7b*. Is this also the case for normal tissue from these same animals? In Fig 3, analysis of expression is focused on tumor RNA/protein. If *Wnt7b* is the major germline susceptibility gene, it might be expected to have an effect on allele-specific expression that is seen before tumors develop, and that increases the probability of tumor initiation. Otherwise it is possible that the allelic differences seen only occur after tumor initiation (possibly in the context of copy number gains on chr 15), and that the polymorphisms in the critical gene(s) are important at late rather than early stages of tumor development. The authors could discuss the possibility that observed allelic differences may only be seen after tumors arise and have begun to progress.

This is now mentioned in the discussion.

Minor points:

1. The resolution of the images in Fig 1 (also Fig 4 and Supp Figure 2) is not very clear and it is not obvious that the staining is really nuclear for B-cat. The scale bar should also be more prominent. It should be possible to provide high resolution images that show the histology in greater detail.

We present the histological pictures with a much higher resolution and include scale bars.

2. In fig 1F the chromosomes labelled 20 and 21 are presumably X and Y
Were any gender differences observed in tumor incidence?

The data in figure 1f are now presented in suppl. figure 3. We have corrected the legend. We did not observe gender differences in tumor incidence or phenotype. This is now mentioned in the text.

3. Fig. 3 suggests that there is preferential allele-specific expression of *Wnt7b* and *Pdgfb* in the tumors, but in the cell lines (Supp Fig 3) for *Pdgfb* the directionality is absent or even reversed. What might be the explanation of this result, given that the cell lines have an apparent gain of the FVB alleles on chr.15 which would be expected to lead to increased expression of the FVB *Pdgfb* allele?? Better discussion why:

Indeed, in the skin tumours both genes are primarily transcribed from the 129P2 allele (now figure 5). The allele preference is less in culture conditions and for *Pdgfb* this preference is even lost (now suppl. figure 4b). We now have compared the absolute transcription levels of *Wnt7b* and *Pdgfb* in culture and in tumors (suppl. figure 4a). The mRNA levels of both genes drop significantly in cell culture but much less for *Wnt7b* than for *Pdgfb* from which transcription is barely detectable. Therefore, the number of *Pdgfb* reads in the allele preference assay is much lower than the number of *Wnt7b* reads and the result might be insignificant. Proliferation of the tumor cells in growth factor-rich culture medium probably does not select for tumor cells with high *Pdgfb* expression whereas it remains dependent on Wnt stimulation. This is in line with the rescue of soft agar colony outgrowth of *Wnt7b* KO tumor cell lines by WNT3a supplementation.

4. Fig legend 3 is very sparse and needs more detail

This figure is now split into two figures (figure 4 and 5). Indeed, the data in figure 3a (now figure 4a) only show that these genes are expressed; we now include control tumours and left out the western blot. Control tumours in figures 4 and 5 represent soft tissue sarcoma's in *Cdkn2ab* deficient mice.

5. Fig 4f has no error bars

Error bars are now included.

6. More information could be provided on the basis for the increased binding at the *Wnt7b* FVB promoter region. Are there polymorphisms that may affect binding of transcription factors in this region? Is there any evidence in favor of the conclusion that increased binding in this region is causally linked to increased expression *Wnt7b*?

The relative increase of H3K27-Ac in this super-enhancer region on the 129P2 allele is in line with the allele preference of *Wnt7b* transcriptional activity. There are multiple polymorphisms in this super-enhancer region also affecting several transcription factor binding motifs. Further investigation is needed to identify the relevant motifs involved in the transcriptional control of *Wnt7b*. For this report we feel this is beyond the scope of this paper.

7. Supp Fig 4. Which genes are in the regions neighboring the peaks in blue? The authors focus on the specific peak upstream from *Wnt7b*, but these other peaks also show allele-specific binding in favor of the FVB allele and could contribute to the phenotypes?

The coverage of the first 2 peaks is rather low and the preference of the third peak is only marginal. Therefore, we put more emphasis on peak 4 which is upstream and close to *Wnt7b*. However, we do not exclude the possibility that the other peaks might represent 129P2 regions (enhancer/promoter) contribute to the (development of the) phenotype as well.

Reviewer #3 (Remarks to the Author):

This study deals with the involvement of Wnt signaling in skin cancer development induced by the loss of *Cdkn2ab* tumor suppressor locus in mice, depending on the genetic background. The authors showed that *Cdkn2ab*^{-/-} mice, generated from 129P2 ES cells, develop TBC-like skin carcinomas but not in the FVB/N background. They found that a 20 Mb region of 129P2 chromosome 15 harboring the *Wnt7b* and *Pdgfb* genes is linked to the skin carcinoma phenotype. *Wnt7b* was shown to be overexpressed in skin carcinoma induced by the loss of *Cdkn2ab* and involved in oncogenic potentials of *Cdkn2ab* deficient cell lines in vitro. Furthermore, the authors identified *Cdk6* as a downstream gene of *Wnt7b* signaling and *Cdk6* caused in vitro transformation of skin tumor cell lines.

These studies are important for the understanding of tumorigenesis involving the somatic deletion of the *Cdkn2ab* tumor suppressor locus. However, phenotypes and findings of this study are observed in mice with specific genetic background and it is unclear whether author's model is consistent with tumorigenesis in human.

Indeed, the phenotypes are observed in mice with a specific background. However, the phenotypic consistency and backcross approach reducing the genetic variation made it possible to identify the relevant genomic region. Our study uncovers a causal relationship between tumor predisposition and natural variation in WNT signaling especially in case of enhanced CDK6 activity which might occur in various human cancers. This makes it worthwhile to further explore how WNT and/or CDK6 plays a critical role in these tumours even in the absence of any mutations in these genes. In support of this we discuss correlations between p15^{Ink4b} loss, CDK6 activation and WNT signaling observed in human cancers.

Major comments:

1. Importance of Wnt7b gene in skin carcinogenesis in Cdkn2ab knockout (KO) mice is not confirmed in vivo. Knockin analysis of Wnt7b in FVB/N mice would support their conclusions. Analysis of the effects of Wnt7b KO on skin carcinogenesis in 129P2 background is also useful. At least subcutaneous and orthotopic tumor models using Cdkn2ab deficient carcinoma cell lines with or without Wnt7b could show the importance of Wnt7b in vivo.

We have addressed the importance of WNT7b in vivo by s.c. transplantation of tumor cell lines in immune deficient mice. However, this procedure did not result in tumor formation which could be due to several reasons e.g. the immune status of the host, the s.c. micro-environment or the lack of heterogeneity.

2. It is not clear that single Cdkn2a KO cells do not respond to Wnt3a in the cell proliferation and colony formation assays (Fig.4b and c). The authors state that upregulation of Cdk6 activity rather than its amount induced by the complete loss of Cdkn2ab locus is important for Wnt-induced cellular transformation. To support their conclusion, the expression levels of Cdk6 in Cdkn2a KO cells (MEFs) treated with or without Wnt3a should be shown compared with those of Cdkn2ab KO cells. In addition, the phosphorylation level of the substrate for Cdk6, such as Rb, needs to be shown to assess the Cdk6 activity in the same condition.

Both in Cdkn2a KO and Cdkn2ab KO MEFs Cdk6 transcription is not induced by WNT3a supplementation whereas Ccd1 is (suppl. figure 14a,b). This will lead to enhanced CDK6 activity but this will be limited by p15^{Ink4b} in Cdkn2a KO MEFs.

To test whether transformation is dependent on pRB phosphorylation per se Cdkn2ab KO MEFs were transduced with dominant active CDK4-R24C. However, this did not result in enhanced colony formation capacity whereas transduction of CDK6 did (figure 6h). Therefore, we believe that CDK6 has a specific role in cellular transformation distinct from CDK4.

3. In Fig. 4f, the amount of β -catenin or the expression of Axin2 or SP5, Wnt target genes, should be shown to assess the change of endogenous Wnt/ β -catenin signaling activity.

We now show that expression of the WNT target *Axin2* is lost upon inactivating the Wnt7b gene and that this can be rescued by WNT3a supplementation in tumor cell lines (suppl. figure 11).

4. Statistical analyses are lacking in this study. Statistical significance should be shown in the individual data.

Statistical analysis of the data is given in each panel when appropriate.

Minor comments:

1. Label and explain the axes in Fig.1e, 1f, and 3a-3c.

We have clarified the axis of the relevant figures (figure 1e and 1f is now suppl. figure 2b respectively; figure 3c is now figure 5a).

2. Fig.3a does not have a control group.

We now included controls in this figure (now figure 4a) and left out the western blots.

3. In page 6, line 12, "WNT antagonist DKK3" is an inappropriate description. It is reported that DKK3 does not have Wnt antagonist activity.

Indeed, somewhat controversial data have been reported about the DKK3 activity in WNT signaling control. Therefore, we changed the wording in 'DKK3, which is implicated in WNT signaling control'.

Reviewers' comments:

Reviewer #1 (Remarks to the Author):

The authors have address my concerns.

Reviewer #2 (Remarks to the Author):

The authors have addressed most of my points that were raised in the first review of this manuscript, including the quality and description of the Figures, and several issues related to data interpretation. I believe that the main critiques have been addressed satisfactorily, and while there are many questions that remain to be resolved, the manuscript is now acceptable for publication.

Reviewer #3 (Remarks to the Author):

This reviewer appreciates the authors' study investigating effects of different genetic background on mouse tumor formation. My concern is that the authors did not show the importance of Wnt7b in skin carcinogenesis using in vivo tumor formation assay. Although the authors showed that Wnt7b is required for the activation of the β -catenin pathway in vitro, but they did not add the data that Wnt7b is important in subcutaneous tumor formation in immune deficient mice. The authors responded to other comments.

Rebuttal

Response to first major concern Reviewer #3

We noticed that the three reviewers indicated that we addressed all the questions and concerns they raised appropriately except for the first major concern of the third reviewer, being the lack of formal *in vivo* proof for the causal role of Wnt7b in the development of TBCs. In this appeal we wish to address this remaining unresolved issue of which the relevance we acknowledge.

As suggested, we performed graft experiments of early passage tumor-derived cell lines in immunodeficient NSG mice. Unfortunately, these grafts did not result in tumor outgrowth as is often observed with primary tumors making it impossible to confirm the effects of Wnt7b in this additional *in vivo* setting. However, our data are supported by a number of published observations made both in mouse models and in a clinical setting.

First, Wnt7b is upregulated in hfSCs during physiological and precocious anagen after BMP inhibition and Wnt7b conditional inactivation during HF morphogenesis impairs HF cycling resulting in a shorter anagen, premature catagen onset with overall shorter hair production. These ‘*in vivo*’ observations strongly support a growth promoting effect of natural Wnt7b overexpression on hfSCs and thereby its oncogenic role in the development TBCs.

Second, clinical data also implicate a contribution of subtle modulation of Wnt7b expression to the development of oral and esophageal squamous carcinoma.

We now have mentioned these published *in vivo* data in the discussion of the manuscript to further substantiate the oncogenic role of Wnt7b overexpression in TBC development.